# Disparities in the prevalence of screened depression at different altitudes in Peru: A retrospective analysis of the ENDES 2019

**Cynthia Alejandra Zegarra-Rodríguez[1,2], Nahún Raphael Plasencia-Dueñas[1,2], Virgilio E. Failoc-Rojas[3]***

**1** School of Medicine, Universidad Nacional Pedro Ruiz Gallo, Lambayeque, Peru, **2** Sociedad Científica de Estudiantes de Medicina de la Universidad Nacional Pedro Ruiz Gallo (SOCIEM-UNPRG), Lambayeque, Peru, **3** Universidad San Ignacio de Loyola, Lima, Peru

\* virgiliofr@gmail.com

**Data Availability Statement:** http://iinei.inei.gob.pe/microdatos/.

**Funding:** The authors received no specific funding for this work.

## Abstract

### Introduction

Depression is a public health concern, nearing 1.5 million cases and accounting for 9.7% of years lost due to disability. Several factors, including altitude, contribute to its development. Altitude has become a topic for recent research, but its association with depressive symptoms has not been fully clarified. Therefore, this study aimed to determine the association between altitude and depressive symptoms in the Peruvian population.

### Methods

A retrospective, cross-sectional study of the 2019 Demographic and Family Health Survey (ENDES in Spanish) was conducted. The dependent variable, depressive symptoms, was measured using the Patient Health Questionnaire (PHQ-9) and the independent variable, altitude, was categorized into: <1500 meters above sea level (masl), 1500–2499 masl and ≥2500 masl. To evaluate the association between altitude and depressive symptoms, we used Poisson regression model, constructing crude and multiple models.

### Results

Of those living at 1500 to 2499 masl and ≥2500 masl, 7.23% and 7.12% had depressive symptoms, respectively. After adjusting for confounding variables, high altitude was found to be associated with depressive symptoms (prevalence ratio adjusted (aPR): 1.38, 95% confidence interval: 1.04–1.84; aPR 1.41, 95% CI: 1.20–1.66).

### Conclusions

A statistically significant association was found between high altitude and depressive symptoms. This may be attributable to hypobaric hypoxia that occurs at high altitudes and its effects on brain function. This study's findings should be considered to identify the population at risk and expand the coverage of preventive and therapeutic measures in high-altitude areas of Peru with poor access to health services.

**Competing interests:** The authors have declared that no competing interests exist.

# Introduction

Depression, one of the primary diseases in the 21st century with more than 4% prevalence in America [1], is a mental disorder characterized by persistent sadness, low self-esteem, and loss of interest in usually enjoyable activities that affects women more frequently than men. In severe cases, depression results in suicide and represents the second leading cause of death in the young population [2]. In Peru, the number of patients with depression is around 1.5 million and represents 9.7% of the total number of years lost due to disability [1].

The development of depression symptoms in an individual is related to various psychological, biological, genetic, and environmental factors [3]. Of these, high altitude is a recently investigated factor. A deficit of oxygen supply to tissues due to the decrease in the partial pressure of ambient oxygen (PaO2) at high altitudes (hypobaric hypoxia) may lead to alterations at the cerebral level, resulting in the development of depression [4]. There have been two promising mechanisms that explain the effect of hypobaric hypoxia on depression. The first one is related to reduced levels of serotonin due to inefficient tryptophan hydroxylase 2 activity, an oxygen-dependent enzyme. The second mechanism is the impaired brain bioenergetics due to altered phosphocreatine system, which fails to deliver high-energy phosphate for ATP synthesis and subsequent normal neuron activity in the brain [4]. It is also thought that mechanisms of adaptations to hypoxia, such as metabolic responses, inflammation, and the activation of chemosensitive brain regions, modulate stress-related pathways associated with depression [5].

The association of altitude with depression symptoms has been supported by studies in animal models subjected to simulated chronic hypobaric hypoxia, observing higher depressive symptoms in female rats [6], and studies that found a higher prevalence of depressive symptoms in residents of high altitudes compared to those residing at altitudes closer to sea level [7].

However, a study at >3500 meters above sea level (masl) conducted with Tibetan and Andean inhabitants found a low prevalence of depression in this population, suggesting that there may be other influential sociocultural factors, such as religion and socio-familial relationships, that may even inhibit the development of depression in the population residing at a high altitude [8]. Furthermore, it is thought that high altitude may reduce oxygen flow to the fetus due to reduced uterine blood flow, which could cause impaired neurodevelopment during early childhood [9]. Infant neurodevelopment may also be altered by insufficient nutrition and iron deficiency in high altitude settings [9].

The existence of an association between living at high altitudes and developing depressive disorders may be attributed to the chronic hypoxia individuals experience when living at high altitude regions, which is related to the reduced efficiency of serotonin synthesis. However, the relationship is not fully clarified; therefore, further studies are warranted in this field. This would favor the identification of at-risk populations that require better access to mental health care to improve the prevention and treatment of depressive disorders in our country.

Therefore, this study aimed to determine the association between high altitude and depressive symptoms in the Peruvian population. We hypothesized that individuals living at high altitude cities are more likely to experience depressive symptoms than individuals living at lower altitude.

# Methods

## Study design and population

A retrospective, cross-sectional study was conducted among the Peruvian general population using secondary data from the 2019 Demographic and Family Health Survey (ENDES in

Spanish) conducted by the National Institute of Statistics and Informatics (INEI). The ENDES has nationwide, regional, and urban-rural representativeness.

This survey has a probabilistic, two-stage (selection of clusters followed by selection of dwellings), stratified, balanced, and independent sampling method based on departments and areas (urban–rural) of all Peru. The sample size estimated for the survey was 36760 dwellings (14760 belonging to capitals of departments and 43 districts from Lima Province, 9340 belonging to the remaining urban areas, and 12660 belonging to the rural areas). More detail about the ENDES methodology is available in the public technical report [10].

The selection criteria of clusters were as follows: The totals of the equilibrium variables were calculated at the cluster level and for the substratum. The number of clusters to be selected in each department was calculated as the division of the expected sample size in the department by the expected average sample size within the cluster. The total estimated sample in each department was distributed proportionally among its substrata (headquarters, urban and rural rest) according to the census population, and within each substratum, the clusters were ordered according to the geographic continuity in serpentine. For each substratum, a list of clusters was prepared with their corresponding population totals (dwellings) and their respective partial population accumulations. In each substratum, the required number of clusters was selected (systematically and with probability proportional to their population size). Fifty balanced samples were generated for each of the 250 strata. This was obtained by combining the 26 departments, the 3 domains and the 6 sociodemographic strata, considering that some departments have neither the 3 domains nor the 6 strata. The sample was obtained by applying the balanced selection method, using as input information the inclusion probabilities, the totals of the equilibrium variables for the stratum and the sample size determined by sample affixation. Finally, it was verified that the number of clusters selected per stratum coincided with the affixation clusters. More detail in https://www.inei.gob.pe/media/MenuRecursivo/publicaciones_digitales/Est/Endes2019/.

The process of selection of participants is shown in Fig 1. The master sample included 3254 clusters, with a total of 36745 randomly selected dwellings. Of them, only 35522 dwellings were surveyed (dwelling response rate of 96.4%). A total of 37474 households were eligible and only 36251 were surveyed (household response rate of 96.5%). The ENDES includes 3 questionnaires. In this study, the health questionnaire applicable to the population ≥15 years of age and the household questionnaire were used [10]. Accordingly, we included people aged

## Study population / Sample

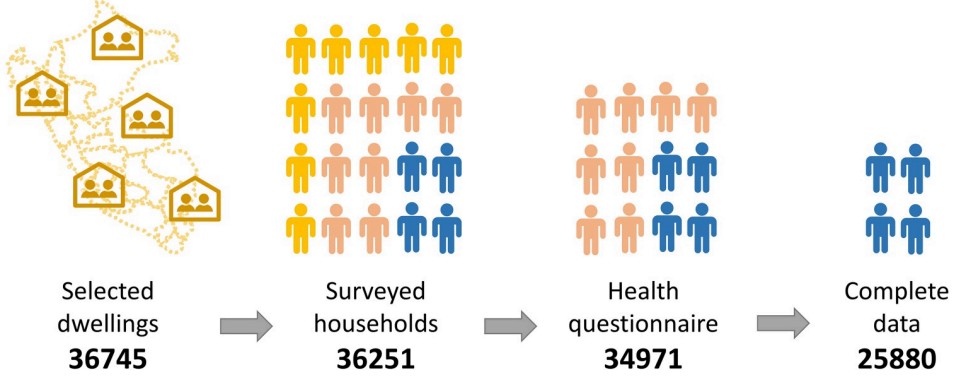

**Fig 1. Selection process of participants.**

15 years or more, who are residents of urban and rural areas of the country, and who responded to the Patient Health Questionnaire (PHQ-9). The individuals with missing and biologically non-plausible data in the independent and dependent variables were excluded. Following these criteria, 34971 residents over 15 years old answered the health questionnaire, but only 25880 had completed the information of interest for the study, resulting in the final sample for analysis.

## Study variables

The dependent variable of our analysis was the presence of depressive symptoms during a period of 14 days preceding the survey as measured by the Patient Health Questionnaire (PHQ-9), which is used for detecting depressive symptoms and determining their severity. The use of the PHQ-9 in the general population and in our country has been validated by experts [11,12]. The questionnaire consists of 9 questions and scores, each one based on severity, starting from 0 (not at all) to 3 (almost every day). According to their cut-off points, they are classified into 5 categories: no depressive symptoms (0–4), mild depressive symptoms (5–9), moderate depressive symptoms (10–14), moderately severe depressive symptoms (15–19) and severe depressive symptoms (20–27). We used a validated cut-off point to classify this variable, which has a sensitivity and specificity of 88% [13,14], and dichotomized the results as not having depressive symptoms (<10) and having depressive symptoms (≥10).

The independent variable was altitude, measured in masl, and was grouped according to the study of Barry & Pollard [15] and a Previous report in Peru [16] into three categories: <1500 (low altitude), 1500–2499 (intermediate altitude), and ≥2500 (high altitude). We have also tried different classifications for analysis: 1) Low (<2500) and high (≥2500) altitude, and 2) low (<1500), moderate (1500–2500), high (2500–3500), and very high altitude (≥3500), as reported elsewhere [17] (see S1 and S2 Tables).

Other variables included based on previous literature were sex (female, male), age (continuous), area of residence (urban, rural), ethnicity (composite variable based on mother tongue and self-reported race: Quechua, Aimara, Amazonian, Afro-Peruvian, White, and Mixed race), educational level (None/Elementary, Primary, Secondary, and Higher), body mass index (BMI), health insurance affiliation (no, yes), hypertension (no, yes), diabetes (no, yes), smoking status (nonsmoker if the person had not smoked a cigarette in the last year, occasional smoker if the person smoked but not daily, and daily smoker if the person smoked at least 1 cigarette every day), wealth quintile (from first to fifth quintile, higher quintile indicating higher wealth level), and disability (dichotomized as yes or no, based on at least one positive response to one of the six disability questions for seeing, hearing, speaking, moving, understanding, or relating to others).

## Data analysis

Stata 16.1 (Stata Corporation, College Station, TX, USA) was used for the analysis. Sampling patterns were analyzed according to the expansion strata and primary sampling unit factors using the *svyset* command.

Population characteristics were described using absolute frequencies and weighted proportions according to the sampling design for categorical variables and mean and standard deviation for quantitative variables. The bivariate analysis was carried out using the weighted Rao-Scott chi-squared test (categorical variables) or Student's t-test (numerical variables), and a p-value <0.05 was considered statistically significant. To measure the association between altitude and depressive symptoms, we used a Poisson regression model and estimated prevalence ratio (PR) with 95% confidence intervals (95% CI), constructing crude and multiple models.

We performed collinearity diagnostics on the independent variables using the *collin* command in Stata, which includes the following tests: variance inflation factor (VIF), tolerance, eigenvalues, condition index, and R-squared (package available from https://stats.oarc.ucla.edu/stat/stata/ado/analysis). For multiple models, we studied the variables that showed statistically significant differences in the bivariate analysis and confounding variables based on previous literature. For sensitivity analysis, we proposed two additional multiple regression models using the exposure (altitude in masl) with two different categorizations based on previous literature.

### Ethical aspects

The ENDES 2019, is a database that is freely available on the INEI website (http://iinei.inei.gob.pe/microdatos/). This database does not contain any information that could reveal the identity of individuals, and the authors cannot identify any participants. Potential participants had to give their consent to participate, and this survey did not involve biological samples and participants were free not to answer any questions they felt uncomfortable with.

### Results

A total of 25,880 persons were included. The weighted proportion of women was 54.23%. The mean age was 36.92 years (SD 13.62), 46.36% had secondary education, 83.23% lived in urban areas, and 72.92% had medical insurance. In terms of habits, 1.53% were daily smokers, and in terms of medical history, 7.36% had high blood pressure, 2.99% had diabetes, 24.40% were obese, and 1.15% had some disability. A total of 75.99% lived at an altitude of <1500 masl, 5.75% at 1500–2499 masl, and 18.26% above 2500 masl. The overall prevalence of depressive symptoms was 5.88% (95% CI: 5.42–6.37) (Table 1).

The proportions of sex, age, educational level, health insurance, hypertension, diabetes, smoking habits, and disability were similar at each altitudinal level. On the other hand, area of residence, ethnicity, BMI, and wealth quintile showed some variation in each altitude category. Because the intrinsic factors were similar in the 3 altitude categories, they were compared in terms of depressive symptoms in bivariate and multiple regression analysis (S3 Table).

Depressive symptoms were significantly more common among women (p<0.001), Aymara people (p<0.001), those with no elementary education or no education at all (p<0.001), those with hypertension (p<0.001) and diabetes (p<0.001), daily smokers (p = 0.002), and those with a disability (p<0.001). With respect to altitude, the prevalence of depressive symptoms was higher in people living at 1500–2499 masl (7.23%) and ≥2500 masl (7.12%) compared to those residing at <1500 masl (5.48%), indicating statistically significant differences (p = 0.002) (Table 2).

On multiple regression analysis (adjusted for age, sex, area of residence, educational level, hypertension, diabetes, smoking habits, wealth quintile, and disability), the prevalence of depressive symptoms was higher in the group residing at 1500–2499 masl (aPR = 1.38, 95% CI: 1.04–1.84) and ≥2500 masl (aPR = 1.41, 95% CI: 1.20–1.66) (Fig 2).

Other variables of interest included being a daily smoker (PR = 3.22, 95% CI: 2.08–4.07), being female (PR = 2.83, 95% CI: 2.32–3.45), having a disability (PR = 2.60, 95% CI: 1.71–3.90), and a diagnosis of hypertension (PR = 1.49, 95% CI: 1.20–1.87) and diabetes (PR = 1.45, 95% CI: 1.04–2.04). On the other hand, having a higher education level was associated with a lower prevalence of depressive symptoms (see Table 3). When using different cut-off points for the categorization of altitude, there were no considerable differences with the main classification (see S1 and S2 Tables). Additionally, when stratifying by ethnic group, the PR of depressive symptoms according to altitude did not show significant associations, excepting for Aimara (PR = 3.41, 95% CI: 1.49–7.82) and mixed-race individuals (PR = 1.54, 95% CI: 1.08–

**Table 1. Characteristics of the participants (*n* = 25 880).**

| Characteristics | *n* (%) | 95% CI |
|---|---|---|
| **Sex** | | |
| Male | 10003 (45.77) | 44.78–46.77 |
| Female | 15877 (54.23) | 53.23–55.22 |
| **Age (years)*** | 36.92±13.62 | 36.58–37.26 |
| **Area of residence** | | |
| Urban | 17932 (83.23) | 82.62–83.82 |
| Rural | 7948 (16.77) | 16.18–17.38 |
| **Ethnicity†** | | |
| Quechua | 4125 (14.09) | 13.30–14.92 |
| Aimara | 525 (1.33) | 1.11–1.59 |
| Native or Indigenous Amazonian | 352 (0.69) | 0.52–0.92 |
| Afro-Peruvian | 2563 (13.78) | 13.03–14.57 |
| White | 1411 (8.71) | 8.08–9.39 |
| Mixed race | 10501 (61.40) | 60.30–62.49 |
| **Educational level** | | |
| None/Elementary | 651 (2.17) | 1.93–2.44 |
| Primary | 5238 (16.61) | 15.94–17.30 |
| Secondary | 12071 (46.36) | 45.34–47.39 |
| Higher | 7920 (34.86) | 33.85–35.89 |
| **Body mass index (kg/m$^2$)†** | | |
| Underweight ($<$18.5) | 309 (1.27) | 1.07–1.50 |
| Normal weight (18.5–24.9) | 9164 (34.15) | 33.29–35.02 |
| Overweight (25–29.9) | 10335 (40.18) | 39.27–41.09 |
| Obese ($\geq$30) | 6069 (24.40) | 23.54–25.29 |
| **Health insurance** | | |
| No | 5943 (27.08) | 26.12–28.06 |
| Yes | 19937 (72.92) | 71.94–73.88 |
| **Hypertension†** | | |
| No | 24271 (92.64) | 92.07–93.16 |
| Yes | 1590 (7.36) | 6.84–7.93 |
| **Diabetes†** | | |
| No | 25277 (97.01) | 96.63–97.35 |
| Yes | 588 (2.99) | 2.65–3.37 |
| **Smoking habits** | | |
| Nonsmoker | 21622 (81.25) | 80.38–82.09 |
| Casual smoker | 3946 (17.22) | 16.44–18.03 |
| Daily smoker | 304 (1.53) | 1.28–1.82 |
| **Wealth quintile** | | |
| Q1 | 7293 (16.14) | 15.53–16.77 |
| Q2 | 6826 (21.35) | 20.49–22.24 |
| Q3 | 5083 (21.77) | 20.90–22.66 |
| Q4 | 3823 (20.87) | 19.93–21.83 |
| Q5 | 2855 (19.88) | 18.94–20.85 |
| **Disability** | | |
| No | 25603 (98.85) | 98.63–99.04 |
| Yes | 277 (1.15) | 0.96–1.37 |
| **Altitude (masl)** | | |

(*Continued*)

**Table 1.** (Continued)

| Characteristics | *n* (%) | 95% CI |
|---|---|---|
| <1500 | 16393 (75.99) | 74.98–76.98 |
| 1500–2499 | 2049 (5.75) | 5.10–6.48 |
| ≥2500 | 7438 (18.26) | 17.32–19.23 |
| **Depressive symptoms** | | |
| No | 24409 (94.12) | 93.63–94.58 |
| Yes | 1471 (5.88) | 5.42–6.37 |

* Mean (standard deviation).

† Some variables may sum to less than 25 880 due to missing data.

Masl: Meters above sea level, CI: Confidence interval.

2.18) (S4 Table). All independent variables did not show intercorrelations after performing collinearity diagnostic tests.

## Discussion

### Main findings

In this study, a significant association was found between high altitude and the presence of depressive symptoms in the Peruvian population (aged over 15 years). The prevalence of screened depression was 41% higher in people residing at altitudes above 2500 masl compared to that in the reference group (<1500 masl) and 38% higher in people residing between 1500–2499 masl.

### Comparison with previous studies

The main result is in favor with the proposed hypothesis and supports the previous literature. DelMastro et al. found in 203870 people from the U.S. that the prevalence of at least one episode of major depressive episode per year was correlated with the substate region mean altitude (r = 0.27, p < 0.0001) [18]. In addition, a study by Zaeh et al. showed a higher prevalence of depressive symptoms in people with chronic respiratory disease living at higher altitudes [7]. There are also a large number of studies conducted mainly in the U.S. that evaluates the link between altitude and suicide rates [4]. A report from the National Violent Death Reporting System from 2005 to 2008 found among 35725 completed suicides that altitude of residence was an independent predictor of suicide in people with bipolar disorder, and that individuals with this condition were at higher risk of suicide at the highest mean altitude compared to individuals with unipolar depression, schizophrenia, and anxiety disorders [19]. Data from the World Health Organization indicate at least 800,000 suicide-associated deaths each year, making it the second leading cause of death among people aged 15–29 years [2]. A strong association has been found between altitudes and suicide rates in 2584 counties in the United States [20], and a study investigating the prevalence of comorbidity in suicide victims found that depressive disorders were the most common of them [21]. In a more regional context, a study in Ecuador showed that suicide rates are higher (9 per 100000 inhabitants) in high altitude provinces [22]. The higher prevalence of depression in high-altitude locations could explain why suicide rates are higher in these settings. However, suicide rates may be attributed to the presence of other factors aside from the methodological limitations of the studies, for example, cultural differences, firearm ownership, and substance abuse [4].

Conversely, a study conducted on elderly residents of the Himalayas and the highlands of Peru did not find any significant association between depression and altitude, as despite the

Table 2. Presence of depressive symptoms according to variables of interest.

| Characteristics | Depressive symptoms | | $p^*$ |
|---|---|---|---|
| | **No** | **Yes** | |
| | **% (95% CI)** | **% (95% CI)** | |
| **Sex** | | | |
| Male | 96.91 (96.38–97.37) | 3.09 (2.63–3.62) | |
| Female | 91.76 (90.98–92.49) | 8.24 (7.51–9.02) | <**0.001** |
| **Age (years)†** | 36.62 (36.28–37.00) | 41.71 (40.16–43.26) | <**0.001** |
| **Area of residence** | | | |
| Urban | 94.12(93.54–94.65) | 5.88 (5.35–6.46) | 0.999 |
| Rural | 94.12 (93.40–94.77) | 5.88 (5.23–6.60) | |
| **Ethnicity** | | | |
| Quechua | 91.71 (89.97–93.17) | 8.29 (6.83–10.03) | <**0.001** |
| Aimara | 88.75 (83.28–92.59) | 11.25 (7.41–16.72) | |
| Native or Indigenous Amazonian | 96.17 (88.45–98.80) | 3.83 (1.20–11.55) | |
| Afro-Peruvian | 94.05 (92.55–95.26) | 5.95 (4.74–7.45) | |
| White | 92.42 (89.79–94.41) | 7.58 (5.59–10.21) | |
| Mixed race | 95.39 (94.74–95.96) | 4.61 (4.04–5.26) | |
| **Educational level** | | | |
| None/Elementary | 87.46 (83.47–90.59) | 12.54 (9.41–16.53) | <**0.001** |
| Primary | 90.47 (88.97–91.78) | 9.53 (8.22–11.03) | |
| Secondary | 94.27 (93.57–94.90) | 5.73 (5.10–6.43) | |
| Higher | 96.08 (95.38–96.68) | 3.92 (3.32–4.62) | |
| **Body mass index (kg/m$^2$)** | | | |
| Underweight (<18.5) | 90.21 (83.82–94.24) | 9.79 (5.75–16.18) | 0.105 |
| Normal weight (18.5–24.9) | 94.44 (93.66–95.13) | 5.56 (4.87–6.34) | |
| Overweight (25–29.9) | 94.32 (93.56–95.00) | 5.68 (5.00–6.44) | |
| Obese (≥30) | 93.53 (92.43–94.48) | 6.47 (5.52–7.57) | |
| **Health insurance** | | | |
| No | 94.35 (93.42–95.15) | 5.65 (4.85–6.58) | 0.541 |
| Yes | 94.04 (93.47–94.56) | 5.96 (5.44–6.53) | |
| **Hypertension** | | | |
| No | 94.67 (94.20–95.11) | 5.33 (4.89–5.80) | <**0.001** |
| Yes | 87.38 (84.76–89.61) | 12.62 (10.39–15.24) | |
| **Diabetes** | | | |
| No | 94.33 (93.84–94.78) | 5.67 (5.22–6.16) | <**0.001** |
| Yes | 87.47 (83.02–90.88) | 12.53 (9.12–16.98) | |
| **Smoking habits** | | | |
| Nonsmoker | 94.05 (93.49–94.56) | 5.95 (5.44–6.51) | **0.002** |
| Casual smoker | 95.04 (93.87–96.00) | 4.96 (4.00–6.13) | |
| Daily smoker | 87.69 (80.92–92.29) | 12.31 (7.71–19.08) | |
| **Wealth quintile** | | | |
| Q1 | 94.40 (93.61–95.09) | 5.60 (4.91–6.39) | 0.064 |
| Q2 | 93.75 (92.80–94.58) | 6.25 (5.42–7.20) | |
| Q3 | 93.43 (92.32–94.39) | 6.57 (5.61–7.68) | |
| Q4 | 93.78 (92.36–94.95) | 6.22 (5.05–7.64) | |
| Q5 | 95.41 (94.26–96.34) | 4.59 (3.66–5.74) | |
| **Disability** | | | |
| No | 94.30 (93.81–94.76) | 5.70 (5.24–6.19) | <**0.001** |

(*Continued*)

**Table 2.** (Continued)

| Characteristics | Depressive symptoms | | $p^*$ |
|---|---|---|---|
| | No | Yes | |
| | % (95% CI) | % (95% CI) | |
| Yes | 78.41 (68.21–86.01) | 21.59 (13.99–31.79) | |
| **Altitude (masl)** | | | |
| <1500 | 94.52 (93.9–95.1) | 5.48 (4.93–6.09) | **0.002** |
| 1500–2499 | 92.77 (90.66–94.43) | 7.23 (5.57–9.34) | |
| ≥2500 | 92.88 (92.06–93.63) | 7.12 (6.37–7.94) | |

\* *P* values calculated with the weighted Rao-Scott chi-square test or Student's *t* test.

† Mean.

Bold text indicates a significant p value (p<0.05).

Masl: Meters above sea level, CI: Confidence interval.

high altitude, the prevalence of depressive disorders was low. This result suggested the possibility of the influence of other factors, such as lifestyle, interpersonal relationships, and religious beliefs [8]. It has been proven that an inverse relationship exists between religious beliefs and the prevalence of psychiatric disorders [23]. Another explanation for the findings of this last study may be the generational adaptation to high altitudes among the Tibetan settlers who have been living at high altitudes for about 25,000 years, whereas the Andean settlers have only been living there for a little more than 12,000 years [24,25]. It has also been proven that the adaptation to high-altitude hypoxia in terms of alveolar ventilation, serum hemoglobin levels, exercise tolerance [25], and chronic mountain sickness [26] is much better in the Tibetan population.

Although this study found an association between altitude of residence and screened depression, this difference was not notable between the two categories stated (1500–2499 masl and over 2500 masl). Even so, the confidence intervals did not show a pattern of increase of the frequency of depressive symptoms according to altitude. This can be explained by the categorization itself, which may lose relevant information. However, using data from 2584 U.S. counties, Brenner et al. found a nonlinear relationship between altitude and suicide risk (a considerable increase occurring between 610 and 914 masl), which suggests that other factors may alter the proposed relationship. In this regard, and relative to our study, the similar PRs found between 1500–2499 masl and over 2500 masl may be explained by the presence of cultural differences. Peru has three different geographical areas (coast, highlands, and jungle), and this has influenced in the development of multiple cultures. For example, in the highlands, there is a strong familiar bond and therefore increased rates of social support. Most of the people live in altitudes around 2500–4000 masl, and this type of support may have played an important protective role against depression. However, there are other individual factors that could influence the development of depression, such as physical activity and sleep quality.

## Explanation of the association between altitude and depression

The state of hypoxia found in people residing at high altitudes is explained by a decreased partial pressure of oxygen ($PO_2$) [4,27]. This is directly proportional to atmospheric barometric pressure (BP), representing 21% of the latter. At sea level, BP is 760 mmHg and $PO_2$ is approximately 156 mmHg. However, at higher altitudes, BP will decrease and consequently, so will $PO_2$, resulting in a decrease in inspired oxygen and tissue hypoxia [4]. In the brain, it has been found that arterial $PO_2$ ranges between 20–25 mmHg in low altitude and increases up to 48

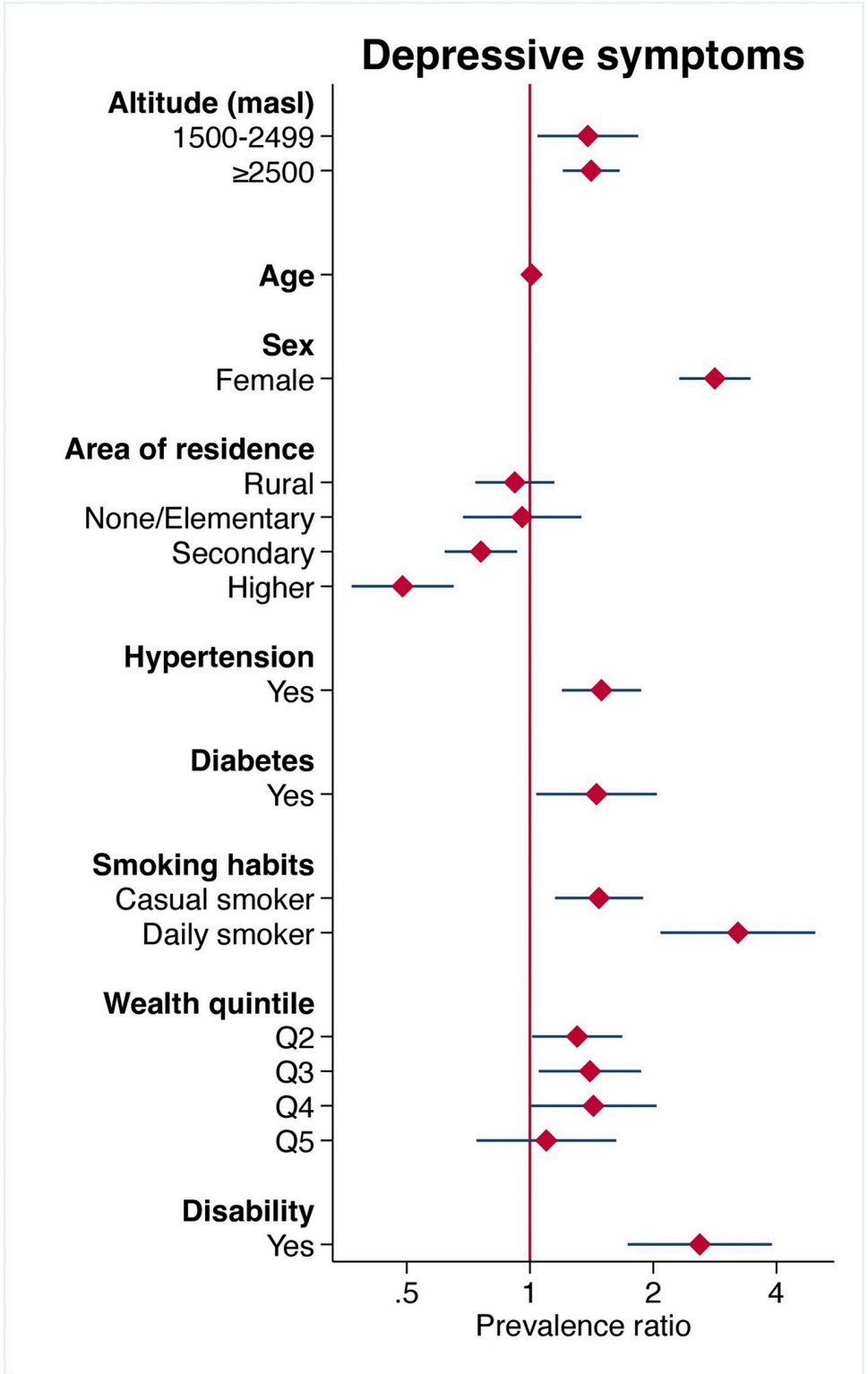

**Fig 2. Forest plot of the adjusted association between altitude of residence and presence of depressive symptoms.**

**Table 3. Regression results on the association between altitude and presence of depressive symptoms.**

| Characteristics | Bivariate analysis | | | Multivariate analysis* | | |
|---|---|---|---|---|---|---|
| | PR | 95% CI | p | PR | 95% CI | p |
| **Altitude (masl)** | | | | | | |
| <1500 | Ref. | | | Ref. | | |
| 1500–2499 | 1.32 | 1.00–1.75 | 0.051 | 1.38 | 1.04–1.84 | **0.024** |
| ≥2500 | 1.29 | 1.11–1.51 | <0.001 | 1.41 | 1.20–1.66 | **<0.001** |
| **Age (years)** | 1.02 | 1.01–1.02 | <0.001 | 1.01 | 1.00–1.02 | **0.002** |
| **Sex** | | | | | | |
| Male | Ref. | | | Ref. | | |
| Female | 2.67 | 2.22–3.21 | <0.001 | 2.83 | 2.32–3.45 | **<0.001** |
| **Area of residence** | | | | | | |
| Urban | Ref. | | | Ref. | | |
| Rural | 1.00 | 0.86–1.16 | 0.999 | 0.92 | 0.74–1.15 | 0.453 |
| **Ethnicity** | | | | | | |
| Mixed race | Ref. | | | - | - | - |
| Quechua | 1.80 | 1.43–2.25 | <0.001 | - | - | - |
| Aimara | 2.44 | 1.59–3.75 | <0.001 | - | - | - |
| Native or indigenous Amazonian | 0.83 | 0.26–2.60 | 0.749 | - | - | - |
| Afro-Peruvian | 1.29 | 1.00–1.66 | 0.046 | - | - | - |
| White | 1.64 | 1.18–2.30 | 0.004 | - | - | - |
| **Educational level** | | | | | | |
| Primary | Ref. | | | Ref. | | |
| None/Elementary | 1.32 | 0.96–1.80 | 0.086 | 0.96 | 0.69–1.34 | 0.797 |
| Secondary | 0.60 | 0.50–0.72 | <0.001 | 0.76 | 0.62–0.93 | **0.008** |
| Higher | 0.41 | 0.33–0.51 | <0.001 | 0.49 | 0.37–0.65 | **<0.001** |
| **Body mass index (kg/m$^2$)** | | | | | | |
| Normal weight (18.5–24.9) | Ref. | | | - | - | - |
| Underweight (<18.5) | 1.76 | 1.03–3.02 | 0.039 | - | - | - |
| Overweight (25–29.9) | 1.02 | 0.86–1.22 | 0.812 | - | - | - |
| Obese (≥30) | 1.16 | 0.96–1.42 | 0.131 | - | - | - |
| **Health insurance** | | | | | | |
| No | Ref. | | | - | - | - |
| Yes | 1.06 | 0.89–1.25 | 0.541 | - | - | - |
| **Hypertension** | | | | | | |
| No | Ref. | | | Ref. | | |
| Yes | 2.37 | 1.94–2.89 | <0.001 | 1.49 | 1.20–1.87 | **<0.001** |
| **Diabetes** | | | | | | |
| No | Ref. | | | Ref. | | |
| Yes | 2.21 | 1.61–3.04 | <0.001 | 1.45 | 1.04–2.04 | **0.030** |
| **Smoking habits** | | | | | | |
| Nonsmoker | Ref. | | | Ref. | | |
| Casual smoker | 0.83 | 0.66–1.05 | 0.125 | 1.47 | 1.15–1.89 | **0.002** |
| Daily smoker | 2.07 | 1.30–3.29 | 0.002 | 3.22 | 2.08–4.97 | **<0.001** |
| **Wealth quintile** | | | | | | |
| Q1 | Ref. | | | | | |
| Q2 | 1.12 | 0.92–1.35 | 0.272 | 1.30 | 1.01–1.68 | **0.040** |
| Q3 | 1.17 | 0.96–1.43 | 0.118 | 1.40 | 1.05–1.87 | **0.021** |
| Q4 | 1.11 | 0.87–1.42 | 0.405 | 1.43 | 1.00–2.04 | **0.048** |

(*Continued*)

**Table 3.** (Continued)

| Characteristics | Bivariate analysis | | | Multivariate analysis* | | |
|---|---|---|---|---|---|---|
| | PR | 95% CI | p | PR | 95% CI | p |
| Q5 | 0.82 | 0.63–1.06 | 0.134 | 1.10 | 0.74–1.62 | 0.646 |
| Disability | | | | | | |
| No | Ref. | | | Ref. | | |
| Yes | 3.79 | 2.48–5.79 | <0.001 | 2.60 | 1.73–3.90 | **<0.001** |

* Adjusted for age, sex, area of residence, educational level, socioeconomic level, hypertension, diabetes, smoking habits, and disability.

Bold text indicates a significant p value (p<0.05).

PR: Prevalence ratio, CI: Confidence interval, Ref.: Reference value.

mmHg in high altitude [27]. The severity of hypoxia will depend on the speed of ascent, the maximum altitude reached, and the susceptibility or adaptation of the individual to high altitudes [24].

However, there is still scarce evidence in humans on how hypoxia affects stress-related pathways associated with depression. Studies in animal models identified depressive behaviors measured by the forced swim test (FST) in rats exposed to hypobaric hypoxia. The FST is a graded test to identify depression-like behavior in rodents [28] and consists of placing the animal in a container of water from which they cannot escape and observing the time elapsed between the moment they begin to swim with the intention of leaving and the moment they become immobile with minimal movement to hold their nose out of the water [29]. After repeated test sessions, the rats adopted immobility much faster, and this is considered as depressive behavior. In these studies, dose-dependent depressive behaviors were found: at higher altitudes, immobility increased and the latency to immobility decreased. This behavior was resolved with the administration of serotonergic antidepressants, but not with noradrenergic antidepressants, which provides evidence of the role of serotonin in hypobaric hypoxia-induced depression [6,29].

In serotonin metabolism, the enzyme tryptophan hydroxylase regulates the rate of serotonin synthesis, using oxygen as a substrate to add it to the serotonin hydroxyl radical. Because of hypoxia, the enzyme tryptophan hydroxylase is not sufficiently saturated with oxygen, leading to inadequate serotonin production [30,31].

Low serotonin levels have been associated with altered mood as well as appetite and suicidal ideation [31,32]. Moreover, decreased serotonin levels have been observed in chronic hypoxic conditions, such as in patients suffering from chronic obstructive pulmonary disease, asthma, or heart disease [31], and in people residing at high altitudes [31,33]. Thus, altitude and its influence on serotonin may be involved in the development of depression.

In addition to the decrease in serotonin, rats exposed to hypoxia were also found to have increased production of metabolites, such as myo-inositol, taurine, and glutamate, which participate in the pathogenesis of several neuropsychiatric disorders, including depression, in the frontal cortex [6,28,29].

## Other factors explaining depression

Other variables whose association with the prevalence of depressive symptoms is worth highlighting in our study are being a daily smoker (PR: 3.22, 95% CI: 2.08–5.00), being female (RP: 2.83, 95% CI: 2.32–2.45) and having a disability (PR: 2.60, 95% CI: 1.73–3.90), which are directly proportional to the presence of depressive symptoms, whereas the degree of education has an inverse relationship with it. These results are consistent with another study in the

Peruvian population, which determined that the factors mostly associated with depressive symptoms were having a disability (OR = 2.40, 95% CI: 1.65–3.49), being female (OR = 2.25, 95% CI: 1.93–2.63), having a history of diabetes (OR = 2.06, 95% CI: 1.49–2.84), and being older than 65 years (OR = 1.91, 95% CI: 1.34–2.72), whereas high levels of education were associated with a low probability of having depressive symptoms [34].

### Relevance for public health

These findings should be considered for the identification of at-risk populations in a current reality in which mental disorders are on the rise. In this way, the coverage of preventive and therapeutic measures can be expanded to high-altitude areas in Peru where access to health services is scarce. In addition, there has been increasing literature on how treatments related to oxygen could prevent the development of depression and other mental disorders, such as hyperbaric oxygen therapy, administration of 5-hydroxytryptophan, or even ventilation techniques for people with acute exposition to altitude [4,5]. Therefore, the information provided in this study supports the need for clinical research on the prevention and treatment of depression in high altitude settings.

### Limitations

This study has some limitations. First, causal relationships between altitude and the presence of depressive symptoms cannot be established in this study as it is a cross-sectional study. Given the nature of the study, some variables that could influence the presence of depressive symptoms (such as religion) were not assessed or had low statistical power (such as violence or alcoholism), which could have resulted in an overestimated association. Finally, there may be a lack of precision and underestimation of frequency due to recall bias or inadequate understanding of questions on disability or depression as the data were self-reported. However, the surveyors were trained, and the PHQ-9 has been validated for use in our country; therefore, we believe that all this should not affect our results.

### Conclusions

In conclusion, this study conducted among the Peruvian general population found a significant association between altitude of residence above 1500 masl and the presence of depressive symptoms. However, the similar result between 1500–2499 masl and 2500 masl does not provide sufficient evidence to state a positive linear relationship. This hypothesis should be tested in future studies including unmeasured physiological, lifestyle, and environmental variables. In this way, the role of hypobaric hypoxia at high altitudes on brain function can be supported by more robust empirical evidence, and ultimately would prevent depression in at-risk populations.

### Supporting information

**S1 Table. Depressive symptoms associated with altitude according to simple ranges.**
(XLSX)

**S2 Table. Depressive symptoms associated with altitude according to extended ranges.**
(XLSX)

**S3 Table. Differences in the presence of depressive symptoms according to variables of interest, stratified by altitude.**
(XLSX)

**S4 Table. Stratification by ethnic group for comparison of adjusted prevalence ratios of depressive symptoms according to altitude.**
(XLSX)

## Author Contributions

**Conceptualization:** Cynthia Alejandra Zegarra-Rodríguez, Nahún Raphael Plasencia-Dueñas, Virgilio E. Failoc-Rojas.

**Data curation:** Cynthia Alejandra Zegarra-Rodríguez, Nahún Raphael Plasencia-Dueñas, Virgilio E. Failoc-Rojas.

**Formal analysis:** Cynthia Alejandra Zegarra-Rodríguez, Nahún Raphael Plasencia-Dueñas, Virgilio E. Failoc-Rojas.

**Writing – original draft:** Cynthia Alejandra Zegarra-Rodríguez, Nahún Raphael Plasencia-Dueñas, Virgilio E. Failoc-Rojas.

**Writing – review & editing:** Cynthia Alejandra Zegarra-Rodríguez, Nahún Raphael Plasencia-Dueñas, Virgilio E. Failoc-Rojas.

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
