## [Decision Letter · Decision Letter 0]

10 Aug 2022

PONE-D-22-16227ALTITUDE AND DEPRESSIVE SYMPTOMS IN THE PERUVIAN POPULATION: ANALYSIS OF A DEMOGRAPHIC AND FAMILY HEALTH SURVEY, 2019PLOS ONE

Dear Dr. Failoc-Rojas,

Thank you for submitting your manuscript to PLOS ONE. After careful consideration, we feel that it has merit but does not fully meet PLOS ONE’s publication criteria as it currently stands. Therefore, we invite you to submit a revised version of the manuscript that addresses the points raised during the review process.

 Please submit your revised manuscript by Sep 24 2022 11:59PM. If you will need more time than this to complete your revisions, please reply to this message or contact the journal office at plosone@plos.org. Please include the following items when submitting your revised manuscript:A rebuttal letter that responds to each point raised by the academic editor and reviewer(s). You should upload this letter as a separate file labeled 'Response to Reviewers'.A marked-up copy of your manuscript that highlights changes made to the original version. You should upload this as a separate file labeled 'Revised Manuscript with Track Changes'.An unmarked version of your revised paper without tracked changes. You should upload this as a separate file labeled 'Manuscript'.

We look forward to receiving your revised manuscript.

Kind regards,

Marcus Tolentino Silva

Academic Editor

PLOS ONE

Journal Requirements:

Additional Editor Comments:

ACADEMIC EDITOR: Please consider all reviewers' suggestions.

Reviewers' comments:

Reviewer's Responses to Questions

**Comments to the Author**

1. Is the manuscript technically sound, and do the data support the conclusions?

Reviewer #1: Partly

Reviewer #2: Yes

2. Has the statistical analysis been performed appropriately and rigorously? 

Reviewer #1: Yes

Reviewer #2: Yes

3. Have the authors made all data underlying the findings in their manuscript fully available?

Reviewer #1: Yes

Reviewer #2: No

4. Is the manuscript presented in an intelligible fashion and written in standard English?

Reviewer #1: Yes

Reviewer #2: Yes

5. Review Comments to the Author

Reviewer #1: This study evaluated a potential role of high-altitude living (Peruvian population) on the prevalence of depression symptoms (using the Patient Health Questionnaire, PHQ-9). The authors report an association between living above 1500 m and symptoms of depression and suggest that hypobaric hypoxia may represent one mechanism explaining this association.

The authors deal with an interesting and important issue from a scientific and a clinical perspective as well. Although this is a well-designed and nicely presented study, several points should be addressed before a final recommendation can be made:

First, the authors may a bit more in detail discuss (in the intro section) why high-altitude living (and hypobaric hypoxia) may provoke depression. Based on that it should be possible to state a clear hypothesis. You may consider (beside others) the following reviews: PMID: 29517615, PMID: 35661753).

Further, the procedure of recruitment (and inclusion) of study participants may be better explained (possibly including a figure), e.g., total persons eligible, inclusion, exclusion criteria, number of respondents, etc.

Are you sure that potential inter-correlations between predictors with have been sufficiently considered by your statistical approach?

The increase in depressive symptoms is not much different from 1500-2500 m to > 2500 m. If hypoxia would represent a major explaining mechanism, one would expect an increase of symptoms with altitude.

There are several other environmental factors that could influence depression,

e.g., UV-radiation, temperature, etc., but also life-style factors (like diet, physical activity, alcohol consumption, etc.) and employment status. You may elaborate a bit on that.

In my opinion, your data do not support the conclusion that the higher the altitude the more frequent are depressive symptoms. Please, check and explain.

A more clinically useful approach should be provided how to implement your findings.

Please, do not forget to mention other (here not or less considered) risk factors like social support, sleep quality/duration, physical activity….

Reviewer #2: Dear Authors of the research paper entitled "ALTITUDE AND DEPRESSIVE SYMPTOMS IN THE PERUVIAN POPULATION:

ANALYSIS OF A DEMOGRAPHIC AND FAMILY HEALTH SURVEY, 2019" with pleasure, I have read your work that seeks to determine the importance of hypoxia concerning the possible effects that this may have on depression (and mood). Your work is interesting and the sample is essential. Having said that, I have a few observations that require your review in order to consider this work as suitable for publication.

General. The title is very unspecific and could be more "interesting" to the reader, for example, I suggest ideas: Analysis of the differences in the depression score (PHQ-9) at different altitudes in Peru" or something like that.

The STROBE reporting format is incomplete, there is missing information and I suggest to review step by step the available template.

Third, it seems to me that a work like this could benefit from a figure showing the differences (pictorially) between the different elevations.

The justification in relation to elevation due to lower PO2 would seem to be very theoretical, there is evidence to suggest that elevation is associated with such findings, among them:

about PO2 and envirment https://www.ncbi.nlm.nih.gov/pmc/articles/PMC6420699/

About suicide at high altitude

https://pubmed.ncbi.nlm.nih.gov/29017474/

About altitude, depression and self-perception

https://pubmed.ncbi.nlm.nih.gov/35020475/

Specific comments

Was the questionnaire conducted in all of Peru? how were the locations classified ? by city altitude ? by canton ? by parishes?

On the other hand, did they use any standard classification about altitude and its ranges? in general there are several classifications (<2,500 m low vs high > 2,500) or the category of low, moderate, high and very high (read this article https://www.frontiersin.org/articles/10.3389/fphys.2021.733928/full )

it is noteworthy that there is less depression above 2500 m and that moderate altitudes have more, so it is essential to categorize the variables on the altitude properly.

In general, the work is interesting, but I think that the observations should be considered to improve the work.

6. PLOS authors have the option to publish the peer review history of their article (what does this mean?). If published, this will include your full peer review and any attached files.

Reviewer #1: No

Reviewer #2: **Yes: **Esteban Ortiz-Prado

---

## [Author Response · Author response to Decision Letter 0]

30 Sep 2022

Response to Journal Requirements

Response: We have ensured our manuscript meets the journal’s requirements.

2. Please clarify the sources of funding (financial or material support) for your study. List the grants or organizations that supported your study, including funding received from your institution.

Response: We have clarified the sources of funding in the cover letter (“The authors received no specific funding for this work”).

3. State what role the funders took in the study. If the funders had no role in your study, please state: “The funders had no role in study design, data collection and analysis, decision to publish, or preparation of the manuscript.”

Response: The study has received no funding.

4. If any authors received a salary from any of your funders, please state which authors and which funders.

Response: No author received a salary from any funder.

5. If you did not receive any funding for this study, please state: “The authors received no specific funding for this work.”

Response: The authors received no specific funding for this work.

Response to Reviewers

Reviewer 1

6. The authors may a bit more in detail discuss (in the intro section) why high-altitude living (and hypobaric hypoxia) may provoke depression. Based on that it should be possible to state a clear hypothesis. You may consider (beside others) the following reviews: PMID: 29517615, PMID: 35661753).

Response: Thank you for your feedback. We have better explained the mechanism of hypobaric hypoxia in the development of depression, including the suggested reviews (lines 54-61).

7. The procedure of recruitment (and inclusion) of study participants may be better explained (possibly including a figure), e.g., total persons eligible, inclusion, exclusion criteria, number of respondents, etc. Are you sure that potential inter-correlations between predictors with have been sufficiently considered by your statistical approach?

Response: Thank you. We have better explained the recruitment and inclusion of study participants (lines 85-123 and Figure 1). In addition, we have assessed the Variance Inflation Factor (VIF) and other collinearity diagnostic tests, showing no intercorrelations between predictors (lines 215-217).

8. The increase in depressive symptoms is not much different from 1500-2500 m to > 2500 m. If hypoxia would represent a major explaining mechanism, one would expect an increase of symptoms with altitude.

Response: Thank you. We have better discussed the role of other factors affecting our results (lines 271-284).

9. There are several other environmental factors that could influence depression, e.g., UV-radiation, temperature, etc., but also life-style factors (like diet, physical activity, alcohol consumption, etc.) and employment status. You may elaborate a bit on that.

Response: Thank you. We have discussed the role of other environmental and life-style factors (lines 253-284).

10. In my opinion, your data do not support the conclusion that the higher the altitude the more frequent are depressive symptoms. Please, check and explain.

Response: Thank you. We have revised the conclusion as suggested.

11. A more clinically useful approach should be provided how to implement your findings.

Response: Thank you. We have provided information on how these findings could be implemented in the clinical context (lines 338-343).

12. Please, do not forget to mention other (here not or less considered) risk factors like social support, sleep quality/duration, physical activity…

Response: Thank you. We have mentioned other important risk factors as suggested (lines 253-284).

Reviewer 2

13. The title is very unspecific and could be more "interesting" to the reader, for example, I suggest ideas: Analysis of the differences in the depression score (PHQ-9) at different altitudes in Peru" or something like that.

Response: Thank you for your feedback. We have modified the title as suggested.

14. The STROBE reporting format is incomplete, there is missing information and I suggest reviewing step by step the available template.

Response: Thank you. We have completed the information using the STROBE reporting guideline.

15. Third, it seems to me that a work like this could benefit from a figure showing the differences (pictorially) between the different elevations.

Response: Thank you. We have added a figure (Figure 2) showing the differences in the prevalence of depressive symptoms according to the altitude.

16. The justification in relation to elevation due to lower PO2 would seem to be very theoretical, there is evidence to suggest that elevation is associated with such findings, among them:

About PO2 and environment:

https://www.ncbi.nlm.nih.gov/pmc/articles/PMC6420699/

About suicide at high altitude:

https://pubmed.ncbi.nlm.nih.gov/29017474/

About altitude, depression and self-perception:

https://pubmed.ncbi.nlm.nih.gov/35020475/

Response: Thank you. We have discussed the evidence on altitude and depression (235-284). We have also added the suggested references in the discussion (lines 251-253 and 286-294).

17. Was the questionnaire conducted in all of Peru? How were the locations classified ? By city altitude ? By canton? By parishes?

Response: Thank you. The questionnaire was conducted in all the regions of Peru. We have clarified/added the requested information in lines 85-110. 

18. On the other hand, did they use any standard classification about altitude and its ranges? in general there are several classifications (<2,500 m low vs high > 2,500) or the category of low, moderate, high and very high (read this article https://www.frontiersin.org/articles/10.3389/fphys.2021.733928/full)

Response: Thank you. We adapted the standard classification on altitude based on the article of Barry & Pollard, as explained in lines 139-141.

19. It is noteworthy that there is less depression above 2500 m and that moderate altitudes have more, so it is essential to categorize the variables on the altitude properly.

Response: Thank you. We have followed the classification of previous literature for the cutoff value of 2500 m (as explained in Comment #18). In addition, we believe this similarity was subtle and the confidence interval is not so precise to interpret an increasing pattern of depression by altitude. The possible reasons for this finding are explained in lines 257-284.

---

## [Decision Letter · Decision Letter 1]

27 Oct 2022

PONE-D-22-16227R1Disparities in the prevalence of screened depression at different altitudes in Peru: A retrospective analysis of the ENDES 2019PLOS ONE

Dear Dr. Failoc-Rojas,

Thank you for submitting your manuscript to PLOS ONE. After careful consideration, we feel that it has merit but does not fully meet PLOS ONE’s publication criteria as it currently stands. Therefore, we invite you to submit a revised version of the manuscript that addresses the points raised during the review process.

ACADEMIC EDITOR: Consider all of Reviewer 2's points. I ask that you take special care to justify the attitude ranges used in population stratification and strictly follow the STROBE guidelines. Please consider writing more about the pathophysiological mechanisms of the findings, as we believe this will be a publication of great interest to our readers. 

We look forward to receiving your revised manuscript.

Kind regards,

Marcus Tolentino Silva

Academic Editor

PLOS ONE

Journal Requirements:

Reviewers' comments:

Reviewer's Responses to Questions

**Comments to the Author**

1. If the authors have adequately addressed your comments raised in a previous round of review and you feel that this manuscript is now acceptable for publication, you may indicate that here to bypass the “Comments to the Author” section, enter your conflict of interest statement in the “Confidential to Editor” section, and submit your "Accept" recommendation.

Reviewer #1: All comments have been addressed

Reviewer #2: (No Response)

2. Is the manuscript technically sound, and do the data support the conclusions?

Reviewer #1: Yes

Reviewer #2: Partly

3. Has the statistical analysis been performed appropriately and rigorously? 

Reviewer #1: Yes

Reviewer #2: Yes

4. Have the authors made all data underlying the findings in their manuscript fully available?

Reviewer #1: Yes

Reviewer #2: No

5. Is the manuscript presented in an intelligible fashion and written in standard English?

Reviewer #1: Yes

Reviewer #2: Yes

6. Review Comments to the Author

Reviewer #1: I appreciate the efforts made by the authors in order to revise their manuscript.

All my points have been considered adequately.

I do not have further comments.

Reviewer #2: Dear authors, thank you for allowing me to review your work entitled "altitude and depressive symptoms in the Peruvian population: analysis of a demographic and family health survey, 2019.

First of all, I would like to congratulate you for bringing to the table such an important issue as the role of the environment, in this case living at high altitudes, on people's health.

Now, once I have reviewed your work, I have many concerns about your manuscript in terms of rigor, that might jeopardize the publication of your work as is, however, I am open to re-consider your work for a future revision if you ammed many of the flaws found within the manuscript.

General comments:

Your paper is not in the correct publication format for this type of study, and you should review the STROBE guidelines and follow each criterion to submit your paper.

Second, their work of revision of the literature is inferior; they have not delved into the subject of the physiopathology of the possible causes behind depression in the inhabitants of high altitudes, nothing is said about the metabolism of serotonin at high altitudes, the role of neurodevelopment is not mentioned, there is no mention of self-perception at high altitude, suicide rates at high altitudes use US data (which has almost no populations living at high altitude) and so on (refeer to the literature for further information https://pubmed.ncbi.nlm.nih.gov/35020475/

Third, the altitude ranges are arbitrary; why do they use that range? Where do they get it from? The altitude ranges have been described previously and are well known among those who do high altitude medicine, they can use the simple range of low and high altitude (< or > 2,500 m) or the range of low (<1500), moderate (1500-2500), high altitude (2500-3500) and very high altitude (3,500 to 4300 m) see the following work for further detail https://pubmed.ncbi.nlm.nih.gov/34675818/

If you cannot analyze the results with those altitude ranges, it is difficult for me to accept it for publication unless you have a substantiated and factual response. Now the statistical analysis is adequate, even though it excludes essential factors such as the length of residence of the highlanders, they were born at high altitudes or they only live at higher altitudes. What about the discussion around culture, why not comparing indigenous people of low altitudes with those indigenouse people at high altitudes, i.e., you can analyze proportions between low versus high, but among the same ethnic group, are there differences?

7. PLOS authors have the option to publish the peer review history of their article (what does this mean?). If published, this will include your full peer review and any attached files.

Reviewer #1: **Yes: **Martin Burtscher

Reviewer #2: **Yes: **Esteban Ortiz-Prado

---

## [Author Response · Author response to Decision Letter 1]

28 Oct 2022

Response to reviewer 2

1. Your paper is not in the correct publication format for this type of study, and you should review the STROBE guidelines and follow each criterion to submit your paper.

Response: Thank you. We have followed the STROBE guideline as suggested. We welcome any additional details that may be necessary.

2. Second, their work of revision of the literature is inferior; they have not delved into the subject of the physiopathology of the possible causes behind depression in the inhabitants of high altitudes, nothing is said about the metabolism of serotonin at high altitudes, the role of neurodevelopment is not mentioned, there is no mention of self-perception at high altitude, suicide rates at high altitudes use US data (which has almost no populations living at high altitude) and so on (refer to the literature for further information https://pubmed.ncbi.nlm.nih.gov/35020475/

Response: Thank you. This information was stated in line 54. However, we have added more detail in lines 71 and 76. 

3. Third, the altitude ranges are arbitrary; why do they use that range? Where do they get it from? The altitude ranges have been described previously and are well known among those who do high altitude medicine, they can use the simple range of low and high altitude (< or > 2,500 m) or the range of low (<1500), moderate (1500-2500), high altitude (2500-3500) and very high altitude (3,500 to 4300 m) see the following work for further detail https://pubmed.ncbi.nlm.nih.gov/34675818/

Response: Thank you for your feedback. We understand your point, but we disagree that the altitude ranges used in the study are arbitrary, as it follows standard classifications (shown in cite 14). Additionally, they were considered in a previous Peruvian report (cite 15) and other articles used similar categorizations (see https://www.ncbi.nlm.nih.gov/pmc/articles/PMC8330906/ and https://www.ncbi.nlm.nih.gov/pmc/articles/PMC8647184/). Therefore, we feel it is of value for the literature. Despite this, we have considered the suggested cut-off points and presented two additional analyses with both simple and extended altitude ranges (Please see Line 149, Line 230, and Supplementary Tables 1 and 2).

4. What about the discussion around culture, why not comparing indigenous people of low altitudes with those indigenous people at high altitudes, i.e., you can analyze proportions between low versus high, but among the same ethnic group, are there differences?

Response: Thank you. We appreciate your comment and contribution to our study. We have analyzed the proportions of the outcome according to the ethnic groups. Please see line 232 and Supplementary Table 4.

---

## [Decision Letter · Decision Letter 2]

25 Nov 2022

Disparities in the prevalence of screened depression at different altitudes in Peru: A retrospective analysis of the ENDES 2019

PONE-D-22-16227R2

Dear Dr. Failoc-Rojas,

We’re pleased to inform you that your manuscript has been judged scientifically suitable for publication and will be formally accepted for publication once it meets all outstanding technical requirements.

Kind regards,

Marcus Tolentino Silva

Academic Editor

PLOS ONE

Additional Editor Comments (optional):

Reviewers' comments:

Reviewer's Responses to Questions

**Comments to the Author**

1. If the authors have adequately addressed your comments raised in a previous round of review and you feel that this manuscript is now acceptable for publication, you may indicate that here to bypass the “Comments to the Author” section, enter your conflict of interest statement in the “Confidential to Editor” section, and submit your "Accept" recommendation.

Reviewer #1: All comments have been addressed

Reviewer #2: All comments have been addressed

2. Is the manuscript technically sound, and do the data support the conclusions?

Reviewer #1: Yes

Reviewer #2: Yes

3. Has the statistical analysis been performed appropriately and rigorously? 

Reviewer #1: Yes

Reviewer #2: Yes

4. Have the authors made all data underlying the findings in their manuscript fully available?

Reviewer #1: Yes

Reviewer #2: No

5. Is the manuscript presented in an intelligible fashion and written in standard English?

Reviewer #1: Yes

Reviewer #2: Yes

6. Review Comments to the Author

Reviewer #1: Thank you.

The authors have already addressed adequately all my comments in revision 1.

I do not have further comments.

Reviewer #2: Dear Authors, the work is much better in this version. The changes are positive. I am still not convinced about the height ranges, and although other people have used them, that does not mean that they are the most recognized, in short, the additional analysis contributes to improving the quality of the work. For the rest, I think the work is fine, I would improve the forest plot somewhat; although the title is good in the graph, it does not sound good, maybe explain it in the title of the figure. For the rest, I have no more comments

7. PLOS authors have the option to publish the peer review history of their article (what does this mean?). If published, this will include your full peer review and any attached files.

Reviewer #1: **Yes: **Martin Burtscher

Reviewer #2: **Yes: **Esteban Ortiz-Prado

---

## [Editor Report · Acceptance letter]

12 Dec 2022

PONE-D-22-16227R2 

Disparities in the prevalence of screened depression at different altitudes in Peru: A retrospective analysis of the ENDES 2019 

Dear Dr. Failoc-Rojas:

I'm pleased to inform you that your manuscript has been deemed suitable for publication in PLOS ONE. Congratulations! Your manuscript is now with our production department. 

Kind regards, 

on behalf of

Dr. Marcus Tolentino Silva 

Academic Editor

PLOS ONE